# Testing and Evaluation of Wi-Fi RTT Ranging Technology for Personal Mobility Applications

**DOI:** 10.3390/s23052829

**Published:** 2023-03-05

**Authors:** Manos Orfanos, Harris Perakis, Vassilis Gikas, Günther Retscher, Thanassis Mpimis, Ioanna Spyropoulou, Vasileia Papathanasopoulou

**Affiliations:** 1School of Rural, Surveying and Geoinformatics Engineering, National Technical University of Athens, 15780 Athens, Greece; 2Department of Geodesy and Geoinformation, TU Wien—Vienna University of Technology, 1040 Vienna, Austria

**Keywords:** indoor positioning, signal-of-opportunity (SoP), wireless-fidelity (Wi-Fi), received signal strength (RSS), round trip time (RTT), fine time measurement (FTM) protocol, ranging assessment, multilateration

## Abstract

The rapid growth in the technological advancements of the smartphone industry has classified contemporary smartphones as a low-cost and high quality indoor positioning tools requiring no additional infrastructure or equipment. In recent years, the fine time measurement (FTM) protocol, achieved through the Wi-Fi round trip time (RTT) observable, available in the most recent models, has gained the interest of many research teams worldwide, especially those concerned with indoor localization problems. However, as the Wi-Fi RTT technology is still new, there is a limited number of studies addressing its potential and limitations relative to the positioning problem. This paper presents an investigation and performance evaluation of Wi-Fi RTT capability with a focus on range quality assessment. A set of experimental tests was carried out, considering 1D and 2D space, operating different smartphone devices at various operational settings and observation conditions. Furthermore, in order to address device-dependent and other type of biases in the raw ranges, alternative correction models were developed and tested. The obtained results indicate that Wi-Fi RTT is a promising technology capable of achieving a meter-level accuracy for ranges both in line-of-sight (LOS) and non-line-of-sight (NLOS) conditions, subject to suitable corrections identification and adaptation. From 1D ranging tests, an average mean absolute error (MAE) of 0.85 m and 1.24 m is achieved, for LOS and NLOS conditions, respectively, for 80% of the validation sample data. In 2D-space ranging tests, an average root mean square error (RMSE) of 1.1m is accomplished across the different devices. Furthermore, the analysis has shown that the selection of the bandwidth and the initiator–responder pair are crucial for the correction model selection, whilst knowledge of the type of operating environment (LOS and/or NLOS) can further contribute to Wi-Fi RTT range performance enhancement.

## 1. Introduction

Knowledge of the position parameters and their variation in time of a person or a moving platform is essential for numerous applications covering all aspects of modern life and human activities. For many years, Global Navigation Satellite Systems (GNSSs) have been widely adopted as the backbone for computing the position solution of a system in motion; however, obstructions to the satellite view indoors and in hybrid environments as well as additional issues (e.g., multipath, atmospheric delays and radio signal interference) render GNSSs incapable of providing an acceptable position solution, and therefore, alternatives and backups are required to efficiently address the localization problem [1,2,3].

Considering target tracking problems, personal mobility applications are of increasingly high interest [4,5], especially following the recent advances in Positioning, Navigation and Timing (PNT) and Internet of Things (IoT) technologies that offer transparent and low-cost localization tools for the general public [6,7]. Cases of pedestrian localization range from open field, outdoor ones to dense urban environments and from indoor, household conditions to deep underground areas. The latter case is by far more demanding, especially when it comes to underground mining and quarry environments in which the provision of ubiquitous, reliable and accurate personnel positioning assures increased safety and productivity KPIs. As GNSSs are incapable of fully or partially supporting the positioning needs in such areas, other approaches (RF-based, inertial, etc.) take the lead in filling this gap [8,9,10,11,12].

User requirements for pedestrian navigation applications are case-dependent. Naturally, while achieving the best possible localization accuracy is always of value, it is also important to achieve a balance between position quality and cost. For instance, for standard personal mobility applications in office environments, it is usually sufficient to achieve corridor-wide accuracies (e.g., 1–2 m) [13]. On the contrary, in other application areas, such as underground mines and quarries, where geometric, material and air quality conditions can be tough, personnel localization is a challenging task requiring the continuous improvement in the technological approach adopted.

In this direction, the recently released Wi-Fi round trip time (RTT) technology has been a substantial advancement as the supporting protocol is available for commercial off-the-shelf (COTS) devices, while more and more Wi-Fi access points (APs) are becoming publicly available as part of onsite infrastructure, offering a promising solution for accurate, low-cost localization of high availability [14]. The published literature on Wi-Fi RTT performance evaluation reveals that raw measurements suffer from instability in various situations, particularly in non-line-of-sight (NLOS) conditions [15,16,17,18], while at the same time a correction model is necessary in order to achieve an acceptable level of accuracy [19,20]. Notwithstanding, many research groups have been studying the potential of the Wi-Fi RTT raw observable for localization, but an exhaustive analysis of its performance characteristics is still not fully available.

In response to this need, this work attempts a full-scale evaluation of Wi-Fi RTT ranging performance as part of a broader research work aiming at providing localization solutions for the underground quarry environments. Wi-Fi RTT range testing is implemented in the form of smartphone-based, indoor pedestrian localization standard-case scenarios. Specifically, investigations involve studying the effects of critical factors in ranging quality including operating bandwidth, sampling rate, operating conditions (LOS/NLOS) and smartphone device type, as suggested in similar approaches [21,22,23,24]. In brief, this paper:(i)Provides an extensive experimental evaluation of Wi-Fi RTT ranging under various conditions and device settings, suggesting optimal setups at certain scenarios;(ii)Investigates and suggests alternative range correction models for removing Wi-Fi RTT biases while explores the use of different mobile devices as initiators;(iii)Analyzes and compares the proposed correction models, as well as the use of different device types concerning their effect on ranging accuracy, and by extension on position fix error.

It is of crucial importance to investigate the ranging capabilities of Wi-Fi RTT and highlight its limitations as well as its potential. Furthermore, through this process, the range-degrading effects that RTT suffers from should be mitigated. Within this scope, the insights gathered via the evaluation can contribute to gaining a better understanding of Wi-Fi ranging behavior and improve the future applications that depend on it.

The structure of this paper is as follows. Following the Introduction, Section 2 provides the state-of-the-art on smartphone-based indoor localization, including key technical details on Wi-Fi RTT ranging. Section 3 presents the assessment methodology and the range correction models adopted in this study. Section 4 discusses data collection scenarios and acquisition, while Section 5 details the results and analyses of the test trials. Finally, Section 6 provides a summary and an outline of this paper.

## 2. Smartphone-Based Indoor Localization

Smartphones have become very powerful devices due to the integration of multi-constellation GNSSs with dual-frequency carrier phases (L1 and L5 in the case of the US Navstar GPS) as well as dual-band Wi-Fi on the 2.4 and 5 GHz frequency. They can be applied for applications ranging from location-based services (LBSs) to simple tasks of applied surveying, which saves time and cost, since no additional hardware has to be purchased and the smartphone is a constant companion anyway. In order to investigate to what extent smartphones are suitable for measurement tasks, the accuracy to be achieved, the measurement effort, the repeatability of the measurement results and the quality of the measurement data are of particular interest.

Especially as dual-band Wi-Fi evolved into a mature technology, its ubiquity became the main driver for smartphone-based localization applications. Outdoor GNSSs remain the main absolute positioning technology due to their high capabilities; however, Wi-Fi can also play a role in GNSS-challenging and -denied environments. In indoor environments in particular, due to the lack of useable GNSS signals, Wi-Fi positioning is capable of serving as an alternative for absolute positioning due to the presence of a large number of Wi-Fi APs in many environments, such as in public buildings, shopping malls, airports, train stations, etc. Although other localization techniques are applicable for indoor positioning (see Section 2.2), Wi-Fi provides inherent advantages, such as its aforementioned ubiquity, which reduces installation costs due to the utilization of already available infrastructures.

In Section 2.1, applicable technologies and techniques for indoor positioning are discussed, followed by a brief discussion of the requirements and capabilities in Section 2.2. Section 2.3 then presents the fundamentals of Wi-Fi positioning and Section 2.4 the usage of RTT measurements. This section highlights the operational principle, the state-of-the art, as well as the potential and limitations of Wi-Fi RTT localization.

### 2.1. Technologies and Techniques

Position determination with the help of smartphones is becoming more and more precise due to recent and fast developments on the sensor market. The aforementioned increasing ubiquity, fueled by the consumer smartphone market, has pushed the need for robust GNSS-like positioning capabilities in GNSS-challenging or -denied and indoor environments. In this section, a brief overview about the type of modern sensors, their state of maturity and adoption is given. A special emphasis is thereby led on inertial navigation and their integration potential together for hybridization with wireless technologies.

#### 2.1.1. Types of Sensors

Several review-type papers in the literature summarize the types of sensors in modern smartphones [25,26,27] that can be used for localization. The Table in [28], published in the Encyclopedia of Geodesy, provides a comprehensive overview of technologies and techniques which can be employed for indoor positioning and navigation. Thereby, technologies and techniques vary depending on the used signals and sensors. Back in 2014, the technologies were classified into three different major system categories in the *Editorial of the Journal of Location-Based Services* by [29]. These classes are: (1) designated technologies based on pre-deployed signal transmission infrastructure, (2) technologies based on so-called “signals-of-opportunity” (SoP) and (3) technologies not based on signals. Infrastructure-based technologies started with the development of systems using infrared or ultrasonic signals, followed by the usage of geomagnetic and/or induced magnetic fields [30], ultra-wide band (UWB) [31] or other RF-based (radio frequency) systems. These wireless technologies are under rapid development also in relation to smartphone localization. The most commonly employed SoP for localization is Wi-Fi [29]. The third category mainly includes sensors for inertial navigation (IN) where relative positioning from a given start location via dead reckoning (DR) can be carried out. Very low-cost sensors based on MEMS (micro-electro mechanical system) technology, such as accelerometers and gyroscopes, are used for this purpose [32]. In addition, vision/camera systems belong to the third category where positioning is performed with scene analysis and visual odometry [33]. Figure 1 summarizes the main technologies and techniques, indicating their state of maturity and adoption.

#### 2.1.2. State of Maturity and Adoption

As can be seen from Figure 1, the PNT (Positioning, Navigation and Timing) ecosystem contains sensors and technologies for relative and absolute localization ranging from a low to high level of adoption and maturity. To name a few important sensors, the first are evidently INS sensors, already providing a high level of adoption and maturity with their integration and fusion with GNSSs and assisted-GNSSs (A-GNSSs). Especially MEMS-based sensor development has revolutionized navigation, together with wireless positioning technologies and their integration into modern smartphones. Their capabilities and performance are also leading to the development of ultra-precise MEMS sensors where a high adoption level can be expected in the near future. As one can expect, multi-constellation GNSSs in smartphones play a decisive role. In this context, PPP–RTK (precise-point positioning together with real-time kinematic) techniques are coming more and more into play. A further performance improvement in this area may also be seen due to future adoption of chip-scale atomic clocks (CSACs). Moreover, smartphone camera systems are also increasingly adopted for visual navigation. On the other hand, the importance of the adoption of ultrasound is, in particular, rather stagnant, although high precision for localization is achievable. The main reason for this may be the high installation costs required if a large environment such as a whole building needs to be covered with receivers and/or transmitters. Ultra-wide band (UWB) technology may take over this role as it can provide high positioning accuracies on the dm-level. The need of infrastructure, however, is still a requirement for UWB systems. Thus, the usage of SoP techniques has a rather high adoption level nowadays for absolute positioning.

#### 2.1.3. Inertial Navigation Philosophy Shift

In the classical navigation approach, INS was employed to bridge gaps of GNSS positioning for a short limited time. Due to the high drift of MEMS sensors resulting in a high error growth, the period to bridge GNSS outages is very short, and frequent updates with known absolute positions are required. Recently, a changed navigation philosophy is applied where INS is considered as a primary navigation sensor. Then, absolute positioning is applied for bounding the INS error growth. This approach allows for a flexible and adaptive blend with other sensors, including unconventional techniques, such as Wi-Fi. Research challenges are then the development and application of flexible software architectures, adaptive data filtering and sensor fusion, stochastic transition between different hybridizations as well as the usage of intelligent algorithms, such as machine learning [35,36,37].

#### 2.1.4. Usage of Raw Sensor Measurements and Data

A further core development in the context of smartphone positioning is their ability to record raw measurements (or also referred to as raw data). As aforementioned, smartphones can receive signals from multi-constellation GNSSs on two frequency bands (L1 and L5, operating at 1575.42 MHz and 1176.45 MHz, respectively) as well as dual-band Wi-Fi on 2.4 and 5 GHz. Wi-Fi ranges can also be considered as raw data leading to the capability to integrate them with ranges from other sources, including GNSS pseudo-ranges in case of their availability. A promising approach will also be their potential capability for integration with UWB ranges. Smartphones of the newest generation will also include UWB. The limitation in this respect, however, is that a UWB infrastructure is still necessary in the environment at dedicated locations (referred to as infrastructure nodes or anchors in a networked solution). Thus, Wi-Fi provides advantages in this respect due to already available infrastructures.

### 2.2. Requirements and Capabilities

#### 2.2.1. User Requirements

In the *GNSS User Technology Report* from 2018 [34], the four main dimensions of PNT systems technology development that enable the future of automated intelligent positioning systems are presented. The key parameters are (1) positioning accuracy; (2) ubiquity; (3) security and (4) connectivity. The following definitions are given:Accuracy is obtained thanks to multi-constellation, multi-frequency GNSSs, augmented by PPP–RTK services and hybridized with INS and other sensors;Ubiquity is provided by complementary positioning technologies and sensors;Security is provided by the combination of independent redundant technologies, cyber-security and authentication;Connectivity relies on the integration with both satellites and terrestrial networks, such as the mobile 5G networks, LEO (low-Earth orbit) satellites or LPWANs (low-power wide-area networks).

Therefore, maintaining performance requires the fusion of multiple positioning technologies and sensors to achieve the goal of continuous ubiquitous navigation.

#### 2.2.2. Smartphone-Based Localization Capabilities

As mentioned in Section 2.1.3, the MEMS-based INS sensors in smartphones can serve as the primary localization technique. Multi-sensor fusion with absolute positioning is a key requirement due to the high INS error growth. Wi-Fi RTT positioning is predestined to estimate absolute positions from the derived ranges via (multi)lateration. The capabilities in terms of accuracies and robust estimation of such derived RTT ranges are the main focus of this study.

### 2.3. Localization Using Wi-Fi

Wireless Fidelity, or for short Wi-Fi (also known as Wireless Local Area Network WLAN) is originally a technology for short-range wireless data communication and is typically deployed as an ad hoc network by attaching a device called access point (AP) in the areas where a wireless Internet access is needed. In the infrastructure topology, APs are the central control point, which forward traffic between terminals of the same cell and bridges traffic to wired LAN. The flexible data communication protocol IEEE 802.11 is implemented to extend or substitute a wired local area network, such as Ethernet. The bandwidth of 802.11 is 11 Mbits and it operates at 2.4 and 5 GHz frequency, which is attractive because it is license-free [38]. For the localization of a mobile device, either cell-based solutions or (tri)lateration and location fingerprinting are commonly employed [28]. Using measurements of the received signal strength indicator (RSSI), a user’s location can be determined. Measured absolute RSSI values are used either directly in fingerprinting or for the RSSI to range conversion using path loss models for lateration [39]. The main disadvantage in RSSI-based Wi-Fi positioning methods, however, is that signal fluctuation and noise as well as various propagation effects on the scanned RSSI values significantly affect the performance. Temporal and spatial variations as well as high signal noise caused by the surrounding environment and its changes usually lead to low achievable positioning accuracies on the several meter levels (see, e.g., [27]). Moreover, signal propagation suffers from server multi-path fading effects due to reflection, refraction, diffraction and absorption by structures and humans. As a result, a transmitted signal can reach a receiver through different paths, each having its own amplitude and phase. These different components are captured by the receiver and an unstable and/or biased version of the transmitted signal is reconstructed. Furthermore, changes in the environmental conditions such as temperature and humidity affect the Wi-Fi signal to a large extent. Consequently, the signal received by a Wi-Fi chip-set at a fixed location varies with time and the physical conditions of the surrounding environment [40]. Furthermore, the presence of people and the user himself affects the localization performance significantly. The signals of APs may even be blocked due to the body of the person being localized in pedestrian positioning applications. The main reason for this is that 2.4 GHz signals can be greatly attenuated by the water in the human body. In addition, the widespread use of Wi-Fi may result in the visibility of hundreds of APs. Since many of these APs are not public, it is not ensured that they are stable. Smartphone hot-spots can even move around. Thus, it is hard to use them in localization algorithms if the environment contains many such APs, while the increase in the number of APs makes this environment even more complex and uncontrollable, which brings several challenges [41].

For these reasons, the round trip time (RTT) and fine time measurement (FTM) protocol has been developed in recent years [14]. Due to the measurement of the two-way time of flight (ToF), referred to as RTT, higher performance for localization can usually be achieved than in RSSI-based lateration. The following section focuses mainly on the IEEE 802.11 mc standard, enabling fine time measurements (FTMs) as well as its operational principle and achievable performance.

### 2.4. Wi-Fi Round Trip Time (RTT)

#### 2.4.1. Operational Principle

The release of the IEEE 802.11 mc standard in 2018 can be seen as a milestone in the development of Wi-Fi localization. The advantage of this standard is that it supports a fine time measurement (FTM) protocol, which allows for the estimation of the distance between a smartphone and an AP using the round trip time (RTT) of the Wi-Fi signal transmission between the two devices. This leads to a significant improvement in the positioning accuracy from several meters as obtained from traditional positioning methods to about 1 m in any line-of-sight (LOS) surrounding environment [14]. To be able to apply the FTM protocol for range measurements, however, several hardware design changes in the existing Wi-Fi chip-sets are necessary for the increase in the timing resolution from the microsecond to the nanosecond level (or even sub-nanosecond level).

The operational principle of RTT FTM is shown in Figure 2. It is a point-to-point (P2P) single-user protocol. For the exchange of multiple message frames between an initiating station (ISTA) sending out the localization request, a responding station (RSTA) is needed for the FTM. Either the smartphone or the AP can be the ISTA and the RSTA is then the other device, respectively. The measurements are carried out in the following steps:The ISTA sends an FTM request to the RSTA;The RSTA receives the request and returns an acknowledgment (ACK) signal to the ISTA;Then, several FTM feedbacks are sent from the RSTA to the ISTA;Then, the mean RTT measurement is used for range calculation.

From these procedures, the total RTT $t_{RTT}$ can be calculated as given in the following Equation (Equation 1):(1)$$\begin{array}{c} t_{RTT}=\frac{1}{N}\left(\sum_{i=1}^{N}t_{4_{i}}-\sum_{i=1}^{N}t_{1_{i}}\right)-\frac{1}{N}\left(\sum_{i=1}^{N}t_{3_{i}}-\sum_{i=1}^{N}t_{2_{i}}\right) \end{array}$$
where $t_{1_{i}}$ is the timestamp when the FTM request is first sent by an ISTA, $t_{2_{i}}$ is the timestamp when the FTM signal arrives at the RSTA, $t_{3_{i}}$ is the timestamp when the RSTA returns the acknowledgment (ACK) signal to the ISTA, $t_{4_{i}}$ is the timestamp when the ACK signal is finally received by the ISTA, *N* is the successful burst number (where *N* > 0, *N* < *B*) and *B* is the total burst number (i.e., burst size, *B* = 8 by selected default).

The protocol excludes the processing time at the ISTA by subtracting $t_{3_{i}}$ − $t_{2_{i}}$ from the total RTT $t_{4_{i}}$ − $t_{1_{i}}$, which represents the time from the instant the FTM message is sent $t_{1_{i}}$ to the instant that the ACK is received $t_{4_{i}}$. This calculation is repeated for each FTM–ACK exchange, and the final RTT is the average over the successful number of FTM–ACK bursts as seen from Equation (Equation 1). The estimated range $r_{est}$ can then be calculated using Equation (Equation 2):(2)$$\begin{array}{c} r_{est}=\frac{1}{2}\times t_{RTT}\times c \end{array}$$
where *c* is the propagation speed of the RF signal.

Ranges to at least three APs have to be measured to obtain a position fix in 2D using lateration [39].

#### 2.4.2. Potential and Limitations

Challenges for Wi-Fi RTT arise in dense-multi-path environments and in NLOS conditions. In such cases, an accurate time–delay estimation may be difficult to achieve as it requires a precise detection of the first signal path with the LOS condition between the two stations and the estimation of its arrival time [42,43]. Our study aims to identify the major influences on Wi-Fi RTT and deriving correction models for performance improvement.

## 3. Assessment Methodology

This section discusses the range evaluation methodology adopted in this paper while it demonstrates the alternative range correction models featured for enhancing the quality of RTT ranging performance.

### 3.1. Operational Aspects and Environmental Concerns

There is a number of parameters to consider affecting the ranging, and by extension the quality of a Wi-Fi RTT position fix solution. For evaluation and assessment purposes, these parameters need to be categorized into internal and external ones in order to investigate their impact on the ranging results. Internal aspects refer to the particular settings and configurations (e.g., sampling rate and bandwidth) related to the technical characteristics of specific sensors. On the other hand, environmental aspects refer to the effects in the context of the influences of the surrounding area (e.g., observation geometry, physical obstructions and radio interference) that may contaminate the signal transmission and penetration behavior [44]. As a result, these factors are treated as influencing elements, and their effect is taken into account at the test scenario planning phase (as seen in Figure 3). The configuration setup and the realization of the test scenarios is discussed in Section 3.2 and Section 4.2, respectively.

As it has already been mentioned, the operational aspects refer to sensor configuration and the device’s technical specifications. In particular, the sampling rate defines the interval between new signal initialization, specifying the amount of gathered data, while the signal bandwidth corresponds to a specific range of transmitting frequencies, providing different levels of stability and accuracy [24]. Another important parameter is the hardware and software differentiation, due to the high variety of smartphone manufacturers.

On the other hand, environmental factors are critical to consider as they are interrelated with the main quality degradation aspect of RF signals, namely signal attenuation and fluctuation. Especially, obstructions in the LOS have a significant impact on the ranging quality [45,46] due to the multi-path and scattering effects leading to signal downgrade. In real-time localization scenarios, the obstruction condition between a transmitter–receiver set is posed as an unknown variable. Analyzing LOS and NLOS data can contribute to ranging quality enhancement by suggesting condition identification schemes [18,47] and then tackle the ranging performance problem, respectively, for each condition.

### 3.2. Wi-Fi RTT Ranging Scenarios

The Wi-Fi RTT raw measurements suffer from an initial distance, position and device-dependent bias which needs to be removed through a correction process in order to achieve more accurate ranges [18,48,49,50]. In this paper, we evaluate the Wi-Fi RTT ranging performance by means of empirically obtained range correction models as well as the effect of the different parameters used. The evaluation process is implemented through a set of observation scenarios based on the parameterization of the quality-degrading factors described in Section 3.1 and in Figure 3.

Experimental testing was undertaken in two stages that create a set of 1D and 2D scenarios. For the first group of test trials, 1D testing examines the effect of operational parameters, namely bandwidth, as well as environmental concerns such as natural obstructions and signal interference. Contrarily, the second group of tests focuses on variations in operational conditions in a 2D space as well as effects on hardware specification. In this regard, user orientation is taken into account as a variable for 2D ranging experiments that define different LOS/NLOS conditions between the transmitter and the receiver. Then, by using appropriate positioning algorithms, the results of 2D ranging can lead to position estimation. The consideration of various alternate bias removal models, based on differently utilized user orientation information, extends the dimensionality of the evaluation process in an attempt to achieve better-ranging estimation and thus better positioning performance.

### 3.3. Correction Models

#### 3.3.1. Bias Removal

The Wi-Fi RTT bias is a systematic error which refers to the difference between the values of the observed range and ground truth range as it originates from the time delay [51]. Moreover, due to reflections and blockages, the signal exposes differently to the expected propagation time, and consequently, it logs a value that deviates from the real one. Bias removal leads to a better range estimation, and thus, more accurate positioning results. It refers to a range correction process, offsetting the logged ranges to more appropriately fit the ground truth values. Such an offset correction value cannot be constant, due to the nature of the bias (distance-dependent), and therefore, a correction model based on raw measurements at known distances/positions could be used instead. The data to obtain the required corrections can be sufficiently represented by a linear fit and can lead to greater accuracies after the execution of the corrections, as shown from previous studies and initial tests [24,52,53].

The correction models rely on the use of datasets employing correction points (CPs), whereas the evaluation of the results obtained on comparisons against selected validation points (VPs) offer independence from the required correction formulae’s parameters.

#### 3.3.2. The Linear Correction Model

A linear correction model represents the best linear fit of a set of values in question. This set of values consists of the raw (measured) ranges and the differences between the measured and the reference ranges. In this way, the model attempts to estimate the optimal linear correlation between the two sets of values. Each measurement corresponds to a correction value which, if applied to the measurement, should reduce, or in the best case scenario even eliminate, the estimated difference between the two values (ground truth distance and the Wi-Fi RTT range).

The formula for the linear model is given by “y=ax+b”, where “*y*” refers to the estimated differences, “*x*” are the raw Wi-Fi RTT ranges, while “*a*” and “*b*” reflect the model parameters. The latter are identified by measurements at reference points of known coordinates (CPs), during the so-called correction phase. Subsequently, with this formula, the optimal correction needed “*y*” for the measured value “*x*” is calculated. At the correction implementation stage, the measured range value is added as follows:(3)$$\begin{array}{c} Correction=a\times Range_{Measured}+b \end{array}$$
(4)$$\begin{array}{c} Range_{Corrected}=Range_{Measured}+Correction \end{array}$$

The linear correction model is used for correcting and evaluating the 1D ranging measurements. Furthermore, it forms the basis for implementing the correction approach for the 2D ranging data. In this case, the user orientation is introduced as a parameter in accordance to Section 3.4.

### 3.4. Alternate Correction Approaches

#### 3.4.1. User Orientation Correction Model

Through adding user orientation as a parameter, datasets of different (but known) user orientations were produced and processed separately. For each dataset, a different linear correction model was produced, tagged at the respective known orientation. Thereby, four cardinal points (N, E, S, W) were used (see Figure 4), leading to four correction models for every AP. These initial linear models constitute the core bias estimation models for use in the sequel of the correction approach. Furthermore, at the initiation stage, each one of the four models was used for correcting the dataset with the corresponding orientation tag. Therefore, each dataset’s ranges would be corrected by using their true orientation model, which in principle should lead to more accurate results.

#### 3.4.2. Multi-Orientation RSSI-Based Correction Model Selection

Considering that user orientation is generally unknown (assuming no external sensor data), the aforementioned approach is not particularly appealing for Wi-Fi-only solutions. This led to the adoption and evaluation of a more advanced model, called multi-orientation RSSI-based (receiver signal strength indicator) correction model. This approach introduces a way of selecting the optimal correction model using data of the four oriented models previously acquired. The algorithm decides which one of the four orientation-based models to use in conjunction with the logged RSS (receiver signal strength) values as shown in Figure 5. In this approach, the RSS values that correspond to the observed ranges are used to identify the most appropriate model for implementing the bias removal process. In practice, the pairs of RSS and range values is modeled using a second-grade polynomial fit. Thereby, each range value refers to an expected RSS value for each one of the calculated oriented models. Then, after obtaining the validation RSS data, the algorithm compares these values with the expected ones from each respective model and selects the one with the minimum absolute difference. This approach suggests that the most suited oriented model should be the one with the most similar RSS profile. At the evaluation stage, the obtained results using this approach can be compared directly with the those from the known orientation approach (Section 3.4.1). Therefore, it serves as a verification of the capability of the method to select the most appropriate model, whilst the level of correlation between the true orientation model and the ranging accuracy is achieved.

#### 3.4.3. Mean Linear Correction Model Selection

The last correction approach employed in this research forms an attempt to bypass the parameter of orientation, by using a standard correction model regardless of the user’s orientation. This model combines all four oriented models in order to produce a new mean correction model, by averaging the defining parameters (*a*, *b*) of their linear models. With the mean model, the special conditions pertained to each orientation affect the raw data quality as well as the bias removal requirements; therefore, they are combined and consequently moderated. In this way, this approach suggests a less complex and more generally applicable solution to the bias problem. These results should be evaluated by comparing them with the aforementioned approaches.

## 4. Data Collection Scenarios and Field Tests

This section provides details of setting up the experimental work in terms of test area selection, hardware/software as well as the planning of individual 1D and 2D ranging scenarios.

### 4.1. Test Area and Equipment

#### 4.1.1. Test Area

The experimental works took place at the lobby and central corridor located at the ground floor of the “Lampadario” building of the School of Rural, Surveying and Geoinformatics Engineering, NTUA. The corridor spans approximately 54 m in total length whilst the adjacent lobby area features free space of approx. 70 m^2^. The corridor offers sufficient length for evaluating ranges up to 50 m long, while the lobby area, including part of the side corridor (see Figure 6), provides an area of about 125 m^2^ for carrying out 2D ranging tests. Figure 7 and Figure 8 show impressions of the test site.

#### 4.1.2. Test Equipment

The test equipment comprises three or five (depending on the experiment) *WILD* Wi-Fi RTT-enabled access points (Figure 9), two Wi-Fi RTT-compatible Android-enabled smartphones, mounting tripods and surveying poles used for secure installation over the accurately surveyed provided positions. The synopsis of the equipment used for each test is detailed in Table 1.

*Compulab Wi-Fi Indoor Location Device (WILD)* (CompuLab Ltd., Yokneam Illit, Israel) APs enable the communication between FTM-compatible Android 9 Pie smartphones supporting the Wi-Fi RTT API utilizing dedicated Android applications. *WILD* is one of the first commercially available Wi-Fi RTT-compatible AP. It relies on the *Compulab fitlet2* platform with an *Intel AC8260* Wi-Fi processor unit. The *WILD* APs were configured at 2.4 GHz band and tested for three different channel bandwidths (20, 40 and 80 MHz) for the needs of this research.

Two Android smartphones are used, i.e., the *Google Pixel 3a XL* (Hon Hai Precision Industry Co., Ltd., New Taipei City, Taiwan) and the *Xiaomi Redmi Note 9 Pro* (Xiaomi Communications Co., Ltd, Beijing, China). Both devices support the 802.11-2016 FTM protocol required to implement the Wi-Fi RTT ranging functionality (both support Wi-Fi 802.11 a/b/g/n/ac). The selection of two devices produced by different manufacturers (despite the limited availability of FTM-compatible devices at the time) was deliberately attempted in order to obtain deeper insight regarding the variations on FTM measurements behavior depending on the manufacturer. The *Google Pixel* operates with Android 9 software, while the *Xiaomi Redmi* operates with Android 10 software.

The open-source Android application *WILD minimal* is selected for compatibility purposes with the *WILD* AP receivers, as it is provided by the same manufacturer. Limited modifications were made to the aforementioned application in order to meet the needs of the experiments. In particular, it was modified to allow for simultaneous measurements recording from several APs in .csv (comma-separated values) files and storage on the smartphone’s internal memory. Moreover, the recording frequency was configured to operate at varying values and thus to enable increased data logging capabilities for case scenarios with rapid motion characteristics (higher dynamics). The parsing of the raw .csv files was conducted using in-house built software in *Python* programming language. The information extracted includes date, time, AP ID, smartphone ID, range, range standard deviation, signal strength (RSS) as well as the count of attempted and successfully recorded measurements.

### 4.2. Test Scenarios Design

#### 4.2.1. Sampling Rate

A crucial operational aspect of indoor range-based positioning systems is the ability to provide sampling rates that suffice the dynamic characteristics of the examined motion. High sampling rates enable both a positioning solution of higher resolution and the ability to more successfully identify and filter out extreme values (outliers) while maintaining ranging information of adequate availability.

Prior to performing the main experimental test trials, preliminary work was undertaken to determine the effect of varying sampling rate on range quality. The evaluated sampling rates vary from 0.5 Hz up to 10 Hz for a fixed reference distance of 2 m between the AP and the smartphone. The resultant ranges, with the exception of an apparent bias, demonstrate minimal variation in average recorded ranges against the nominal accuracy levels, as illustrated in the box plot of Figure 10. In order to facilitate both range logging at faster pace and in order to minimize the test trials’ duration, a sampling rate of 10 Hz is adopted in this study.

#### 4.2.2. 1D Ranging Tests

For the evaluation of RTT ranges in a 1D configuration, the *WILD* APs are placed side by side on a dedicated platform mounted on a tripod while the smartphone is mounted on a geodetic pole at the same level as the APs. AP1, AP2 and AP3 are preconfigured to operate at a 80, 40 and 20 MHz bandwidth, respectively, at a 10 Hz sampling rate.

This experiment includes three scenarios that create three different operational conditions against a reference range varying from 0.5 m up to 50 m. Most of the reference distances are treated as correction points (CPs), whereas a subset of them serves as validation points (VPs). In scenario 1, the APs operate simultaneously at a LOS condition, logging data for approximately 30 s for each trial. In scenario 2, the APs operate simultaneously at NLOS conditions for 30 s, similarly to scenario 1. NLOS condition was achieved by considering the user as an obstacle. In real conditions, users act as true obstacles in most cases. Finally, scenario 3 was executed to evaluate whether the simultaneous operation of APs at a close proximity (0.15 m) between them in scenarios 1 and 2, results in contaminated ranging data, potentially due to interference. As scenario 3 serves mainly as a control for the first two scenarios, it is created only for a subset of the reference ranges while each AP operates separately. Table 2 summarizes all three scenario configurations.

#### 4.2.3. 2D Ranging Tests

The implementation of this experimental setup enables the investigation of Wi-Fi RTT ranging capabilities in realistic conditions using five APs installed at locations that enable positioning operation. In this experiment, an additional smartphone device (*Xiaomi Redmi Note 9 Pro*) was made available to enable the evaluation and comparison among varying hardware. In order to investigate the effect of LOS/NLOS functionality in the 2D setup, range data were collected at four different orientations (N, E, S, W) and at each pre-surveyed point. These points were marked on the floor and determined by using a total station and calculating their local coordinates afterward. As the experimental area consists of two parts, the lobby area and the corridor, it is possible to investigate FTM ranging performance at varying geometries in a controlled manner. As illustrated in Figure 11 and summarized in Table 3, the data collection took place for fourteen correction points (CPs) and four validation points (VPs) at selected locations that cover the test area optimally, also considering the coverage of transition areas by including CP6.

Following the analysis of the 1D ranging experiment (Section 4.2.2), the APs are configured to operate at 80 MHz and at a sampling rate of 10 Hz. The data collection took place at each point and for each orientation for approximately 15 s. Data logging at varying orientations enables the investigation of bearing effect on the resulting range correction in an attempt to optimally mitigate systematic range errors (bias) and range-dependent errors.

## 5. Analysis and Results

The experimental campaigns provided two distinct groups of 1D and 2D range datasets for which the different scenario designs enables the analysis for varying internal and external factors.

### 5.1. 1D Ranging Tests

The histograms gathered in Figure 12 reveal the nature of the raw-range observables both for scenario 1 (left) created in LOS conditions and for scenario 2 (right) in which NLOS conditions are evident for all three bandwidth selections. Three representative reference distances are depicted and demonstrated here (5 m, 15 m and 50 m), where the main variations between APs and the corresponding LOS/NLOS conditions are apparent. Two points are immediately evident: Firstly, NLOS conditions result in range distributions of a high dispersion irrespective of bandwidth setup. Secondly, the 80 MHz setup exhibits a larger systematic range offset (bias) whilst the 20 MHz setup suggests a higher measurement dispersion. Considering the values obtained for the EPDFmax index (suited for describing reliably the prevailing ranges), the corrected measurements quality metrics before and after the linear correction model implementation indicate improved performance for all selected bandwidths and especially for 80 MHz.

Further insights into the behavior of the sensor units under evaluation is offered through the illustrations of Figure 13, in which ranging and correction performance is demonstrated for the complete reference distances set. Specifically, when considering LOS conditions, the ranges demonstrate a similar level of improvement for all bandwidths when a correction model is applied. Contrary, for the case of NLOS data, the 80 MHz setting results in improved range accuracy with minimal outliers, especially for distances up to 40 m, whereas the rest of the APs present higher trueness values starting from reference distances of 15 m. On the other hand, the logged signal strength values are higher for the 40 MHz and 20 MHz settings, indicating a potentially increased measurement availability compared to 80 MHz both for LOS and NLOS conditions.

Following the initial indications regarding ranging quality (repeatability and trueness), the comparison between scenarios 1 and 2 is best demonstrated through Figure 14. The complete datasets are presented with respect to the reference distances, providing insight regarding both measurements deviation as well as trueness. The larger offset is apparent for AP1 whilst AP2 and AP3 are more prone to reporting range outliers.

Figure 15 shows the empirical cumulative distribution comparing the absolute error distance (*x*-axis) for the three APs and the change in their performance in LOS and NLOS conditions. Notably, the results are computed for the 10 validation points that did not contribute to the generation of the correction models. These plots indicate that, despite the varying initial bias, which is visible in the graphs for the original data, when the appropriate correction model is implemented, a similar accuracy improvement can be obtained for all three bandwidth settings (close to 1 m for 80% of the validation data). However, as the CDFs are generated using the EPDF-max ranging values for each VP, the dispersion information is not visible.

The analysis of scenarios 1 and 2 indicate higher performance when utilizing AP1 (80 MHz) since it demonstrates increased stability both for LOS and NLOS conditions, and especially for distances up to 30 m. As the complex nature of interior spaces tends to present multiple obstacles due to walls and equipment, distances exceeding 30 m very rarely offer LOS or even slight NLOS conditions. Moreover, since a higher bandwidth setup is expected to lead to a greater ranging accuracy, while at the same time the larger initial bias may be modeled and mitigated sufficiently, the selection of the 80 MHz setting may present the optimal conditions for the next analysis steps of this study.

As expected, NLOS conditions lead to the contamination of datasets with multi-path-induced errors, resulting in longer range recordings (NLOS-positive bias) as well as higher measurement dispersion that consequently degrades the corresponding ranging quality. Overall, the resulting ranging accuracy after the implementation of correction models lies systematically up to 1 m for 80% of the data sample.

The results of the evaluation regarding the simultaneous and separate APs’ operation are presented for AP1 in Figure 16, where a comparison between scenario 1 and 3 through a combination graph is conducted. The ranging results are quantified by the combined plot of the EPDFmax values of the datasets with the X’X axis presenting scenario 1 and the Y’Y axis presenting scenario 3. In addition, the results for all the APs are summarized in Table 4, achieved via their respective absolute difference between the separate and simultaneous ranging values. The validity of the measurements for scenario 1 is proven by the close proximity of the graph nodes to the y=x line, as well as by the small absolute difference values reported in the table, indicating the normal ranging operation both for simultaneous and separate setups.

### 5.2. 2D Ranging Tests

Following the collection of Wi-Fi RTT ranges for all CPs, the distinct linear correction models are generated for each AP–smartphone pair for the four cardinal orientations. Using these models, the correction multipliers were applied on the observed ranges. Figure 17 presents the dataset collected for VP2 for the ranges between AP1 and the two smartphones for each orientation setting. The preliminary investigation suggests distinct differences in the behaviors between the two smartphones. Regarding the initial ranging offset values, both devices report a total difference in the order of 23 m; specifically, the *Google Pixel* provides ranges in the order of −4 m, whereas the *Xiaomi Redmi* reports ranges in the order of −19 m. Moreover, considering the LOS/NLOS effect, the *Google Pixel* provides more stable ranges with lower standard deviation for the predominantly LOS orientations (North, East and South), as well as higher dispersion and different offset for the NLOS West pointing orientation. On the other hand, the *Xiaomi Redmi* reports ranges of improved quality for North and South LOS orientation, with an increased offset difference of about 2.5 m for West NLOS orientation after correction, whereas the ranges for the East orientation of the ranges dataset demonstrates higher standard deviation than anticipated for the optimal LOS conditions.

Figure 18 presents the range correction models as generated for both units, considering all CPs and orientation settings accordingly with respect to AP1. It is noted that the mean correction model refers to the linear correction models generated utilizing the mean values of all orientations for each CP. Furthermore, the second-order polynomial curves for RSSI values are illustrated.

Since the correction models play a major role in this study, as seen in Table 5 and Table 6, the fitting parameters of the linear models, for the two utilized smartphones, along with the respective standard error (SE) for each oriented dataset are demonstrated. These tables can help distinguish the variation but also the similarity that occurs in the model parameters depending on the different APs and datasets, as well as the performance of each pair. Moreover, in order to also evaluate the RSSI polynomial fitting curves, similar tables for both rovers are provided (see Table 7 and Table 8).

Regarding the efficiency of the proposed *RSSI-based* correction model, the initial evaluation refers to the estimation of the correct orientation selection success. This metric represents the percentage of the correctly assigned measurements to the appropriate correction model based on the reported RSSI values. Table 9 provides the success rate for all VPs as well as the achieved performance represented by the root mean square error (RMSE) and standard deviation (STD) for each orientation. The RMSE KPI indicates a better performance the lower its value. Concerning the *Google Pixel*, the correct model selection appears as being heavily affected by the LOS/NLOS conditions of the device. A clear example of this behavior is presented for the case of VP2 and AP1, where the optimal LOS orientations E and S present high successful selection values, whereas for N and W the percentage drops significantly. When considering the RMSE values, there are cases of correlation between successful model selection and achieved trueness; however, it is not possible to draw a definitive conclusion and this can be mainly attributed to the similarity between models of different orientations, as can be seen in the AP1 example of Figure 18.

Regarding the *Xiaomi Redmi* device *RSSI-based* model selection, a heavily reduced success rate is produced with most cases presenting 0% success (Table 10). This behavior can be mainly attributed to the highly unstable RSSI values produced. In most cases, the RSSI data collected at VPs present a large initial offset affecting the minimum difference from correction models and consequently resulting in a tendency to select the North model that lies closer to the offset values. This is also demonstrated by the increased cases of 100% for the North model when the remaining orientations report 0%. It is also noticeable that there are some “*NaN*” values in the table, corresponding to datasets in which the connection between the smartphone and the respective AP was unsuccessful during the data collection period, leading to not logging any data. This data loss has been observed only for the *Xiaomi* device, and the signal instability is possibly related to the high value of the initial bias, as well as the existing obstructions.

In order to evaluate the different correction model approaches that this paper addresses (as seen in Section 3.4), Figure 19 visualizes via arachnoid diagrams, indicatively, the RMSE values (in m) for both smartphone devices referring to the VP2 datasets. The choice of demonstrating the results from VP2 is due to the optimal position of this validation point, as it has minimum obstructions for all the available APs and a smaller influence of the corridor’s walls in signal transmission. By comparing the results for the different correction models, it is found that the correction with the known *oriented* model generally leads to better results, the *mean* correction model, while it has slightly reduced accuracy than the *oriented* model, it provides improved ranging accuracy compared to the *RSSI-based* method. Additionally, comparing the results from both devices, the *Google Pixel* achieves higher accuracies than the *Xiaomi Redmi* with RMSE values mainly around 1 m, compared to 2 m in certain circumstances, respectively.

Although the ranging error can reach a sub-meter level, it is crucial for this accuracy to be consistent with consequent environmental changes (orientation and obstructions). In this scope, the evaluation takes into consideration the ranging performance in varying conditions in order to suggest the optimal option. Despite the fact that, in difficult situations, there is increased inaccuracy for the *oriented* model, overall, the results indicate that known orientation models for the *Google Pixel* are more accurate, while the results from the *Xiaomi* device and especially with the proposed *RSSI-based* model are more susceptible to errors. The *mean* correction model approach appears to offer a viable solution when user orientation is an unknown variable, but in situations where this information is available through different means (e.g., embedded MEMS IMU), the correction with the appropriate model could lead to optimal results.

As for the *RSSI-based* approach, despite the fact that the concept aims for the algorithm to utilize the RSSI observables to select the best model for correction, depending on previously acquired data, the analysis indicated difficulties in improving the achieved ranging accuracy compared to the other approaches. The reason behind this observation may be attributed to the highly sensitive nature of RSSI observables as well as to the similarity in the model-generated RSSI values driving the algorithm to select a false orientation setting. Although, as previously noted, the “correct” model does not always lead to reduced errors, the reliable selection of that model is a step toward a better implementation of this approach.

## 6. Discussion and Conclusions

### 6.1. Evaluation and Assessment

Section 5 summarized the experimental results obtained for the Wi-Fi RTT ranging scenarios. Specifically, comparisons among different sensor configuration settings, field of view conditions, smartphone devices and range correction models were studied. Furthermore, in order to better illustrate the effect of pointing direction and observation geometry in range accuracy, the comparisons included 1D and 2D test trials. The WiFi RTT ranging functionality assessment is implemented through the comparison between the estimated ranging performance based on reference measurements and the respective nominal specifications provided by the manufacturer. Moreover, respective and relevant analysis in similar environments utilizing alternative ranging technologies has been the subject of previous work from the authors (see [20,21,22,23]), and thus the direct comparison between the technologies is beyond the scope of this paper.

In the 1D test case, range measurements were undertaken along a building corridor at different bandwidth settings as well as at variable LOS and NLOS conditions. Data analysis revealed that the observed ranges are more stable at a 80 MHz bandwidth setting, offering improved performance even in obstruction conditions and despite the excessive value of initial bias encountered. Furthermore, notwithstanding a high deviation and initial bias values occurring for the NLOS case, the implementation of a correction model results in a remarkably improved range accuracy. That said, experimental results indicate that a sub-meter level accuracy is possible to achieve at optimal observation conditions; that figure scales down to 1–1.5 m for a scenario in a typical obstruction environment.

For the 2D case scenarios, range measurements where collected among RPs scattered in a 2D space area featuring the building entrance foyer and a corridor. In this setup, two smartphones were used operating at 80 MHz bandwidth at all APs. Trials were conducted in different user orientations to allow for testing in variant, albeit controlled, LOS/NLOS conditions. In addition, in order to compensate for the lack of orientation information encountered in real-life scenarios (assuming RF-based technologies only), the evaluation of the performance of alternative correction models was also undertaken.

The obtained results suggest that user orientation impacts the quality of the raw-range measurements. Specifically, the relative geometry of the smartphone, the ranging AP and the user’s body location may induce NLOS conditions that potentially alter the signal transmission path between the smartphone device and the AP in question. The collection of range measurements at quadrant orientation settings results in bias mitigation models depending on user orientation. Thereby, the known orientation (*oriented*) model offers improved ranging accuracies, while the *mean* correction model approach achieves similar albeit slightly reduced ranging accuracies. Finally, the proposed *RSSI-based* model selection approach faces significant difficulties on providing the correct orientation model selection.

Concerning the ranging performance observed between the tested smartphone devices, analysis confirms different range biases, and thus, different correction requirements are deemed necessary. The *Xiaomi* measurements indicate a greater deviation from the ground truth ranges as well as increased instability, leading even to a complete dataset loss. The results obtained from the *Google* device are similar to those obtained in the 1D experiments, ascertaining their consistency. Analysis revealed that undertaking further testing with the *Xiaomi Redmi* at different bandwidth selections may provide deeper insights into the device’s ranging capabilities, although no radical improvement is expected in the end results. Finally, the different correction models present similar behavior with the respective smartphone device datasets, with a minor ranging error reduction apparent for the *Google Pixel* device.

The 2D experiments enabled the investigation of the effect of observation geometry thanks to the existence of RPs in the test area (corridor and entrance room). Indeed, the different area settings result in different signal transmission conditions, which in turn impact the ranging measurements. Naturally, the corridor creates obstruction and reflection conditions, resulting in degraded signal transmission and by extension deteriorated ranging performance from the APs deployed in this area. Notwithstanding, this paper focuses on Wi-Fi RTT range assessment, and the elongated corridor geometry visibly impacts the position fixing accuracy due to the APs’ challenging deployment geometry and consequently the performance of the trilateration algorithms.

### 6.2. Toward Pedestrian Localization—Future Steps

Having tested and evaluated the performance of the different Wi-Fi RTT range correction models as well as different devices, it is of interest to also investigate their potential positioning capabilities. The presented position fixing results are generated through a multilateration algorithm and are bound to their particular limitations and issues. Multilateration is highly affected by ranging quality (extensively analyzed in Section 5) and APs geometry deployment. In the frame of this study, we performed a preliminary estimation of the static position of the VPs using the corrected ranges from the 2D experiments. The main focus is the evaluation on the effects of the ranging measurements when utilized for localization as well as discussing the additional position quality-degrading parameters that derive from the employed positioning technique. The use of the complete validation ranging datasets, grouped by logged time from three to five available APs per epoch, allows for a more detailed evaluation of the discrete localization points instead of an average value for each VP.

Figure 20 shows that both the true position of the VP (in relation to the AP positions) and user orientation impacts the localization results. In addition, from the previously discussed ranging errors (see Section 5.2), it appears that in the corridor area the dispersion in the *y*-axis (across-side) is larger compared to the one in the *x*-axis error (along-track). This is due to the challenging (elongated) corridor geometry, leading to more significant errors perpendicularly to the corridor’s axis. It is also noted that for VP2, which was previously addressed as the one with optimal location, the point cloud is rather small, especially for the three orientations that grant LOS (N, E and S). This finding directly relates to the degrading effect on ranges due to obstructions that by extension affect position fix quality. The problems encountered with the *Xiaomi* unit are rather clear in static positioning scenarios indicated by the larger scattering of point cloud position solutions and the degradation of position trueness overall. As expected, this effect is particularly evident in the corridor area wherein observation geometry conditions are less favorable. Although the visualization of the positioning results’ accuracy could be achieved through error ellipses adding dimensionality in the figure, for the sake of simplicity, at this preliminary stage, positioning analysis is considered redundant, as well as the point cloud demonstration, which serves the authors’ purposes best. Moreover, the adoption of a more complex positioning algorithm is considered, which extends the context of this manuscript beyond its scope and is not addressed further.

### 6.3. Use Case Characteristics Affecting Solution Selection

The suitability of the different devices and methods depends on the prevailing conditions and requirements, a substantial number of which vary in different use cases. The conditions affecting the localization performance can be classified under the different categories: built environment characteristics and pedestrian density, while the localization performance requirements are determined through the interaction type, as well as the type and criticality of the operation that utilizes it.

Indoor facilities, where applications utilizing pedestrian tracking are in use or have the potential to be beneficial for specific operations, range from simple to more complex infrastructures, including public transport stations or terminals, airports, shopping malls, office or other types of buildings (e.g., schools, universities, hospitals) and parking areas [54]. In particular, the indoor parking areas attract additional interest combining both pedestrian and vehicle tracking capabilities [55] leading to various opportunities [56,57]. These environments may display different characteristics from the built environment geometry, affecting localization performance. Examples include corridor/area width dimensions, walls or other obstacles. An additional element involves the existence of different floor levels and the manner in which pedestrians advance from one level to the other, i.e., whether this is performed through stairways, escalators or elevators [58].

Pedestrian density also varies in the different use cases both temporally and spatially. Considering the temporal element, peak and off-peak periods display entirely different density levels. Considering the spatial element areas in the vicinity of check/control areas in public transport stations and terminals, exits or entrances, narrow corridors or locations where bottlenecks appear also demonstrates higher pedestrian density.

Most indoor infrastructures involve pedestrian-only movements, and even in areas where pedestrians co-exist with other transport modes there are exclusive areas for pedestrian movement, thus presenting pedestrian-only interactions. However, at specific facilities such as parking areas or bus terminals, interactions between pedestrians and vehicles also occur. Performance requirements in such cases are higher [59].

Location data can be utilized for different operations indoors while most cases serve pedestrian navigation and wayfinding needs. Other examples include simple pedestrian counts for marketing and demand/supply studies (e.g., number of pedestrians at a bus stop/platform or consumer distribution at mall districts).

There exist, however, more demanding applications such as those servicing collision avoidance scenarios indoors [59]. Especially in difficult environments such as underground quarries, user requirements become more stringent for safety purposes.

Last, the criticality level affects the performance requirements. This depends on whether the application (e.g., navigation) operates under normal versus under emergency conditions. For example, the accuracy of the navigation is not as important under normal conditions, while in the case of emergency evacuation it is of vital importance. In indoor environments, emergency measures during disasters necessitate accurate planning and prevention to be able to respond instantly and effectively and evacuate or relocate people [60]. The same applies for pedestrian counts under normal operations or during evacuation actions. Another dimension related to criticality involves the user category. Navigation is more challenging and its performance requirements are higher for specific user categories such as visually or mobility impaired people, children, elderly, and so on, compared to healthy adults [61,62].

### 6.4. Concluding Remarks

High-quality (accuracy, trueness, availability) position estimation is the ultimate goal for many indoor mobility applications. This goal is particularly difficult to achieve in areas featuring challenging observation conditions (obstructions of various types, narrow pathways, etc.) that downgrade both ranging and localization quality. Particularly, considering pedestrian localization, with the exception of static position fixing, it is critical to efficiently track the user location and kinematics in near-real time and high update rate. Indeed, for the case of RF-based systems, this requirement asks for accurate and continuous range estimation, exempted of outliers and unstable measurements, in order to compute user location in a precise and robust manner.

In this regard, a thorough characterization of Wi-Fi RTT ranging performance presented in this study sets the basis for pedestrian localization using this technology. However, with the exception of range quality, position fixing depends heavily on user dynamics, update rate of observations and the localization algorithm encountered. Therefore, due to its complicated character, a detailed investigation of the localization problem is beyond the scope of this study.

Nevertheless, preliminary testing confirms the importance of range correction models in position fixing. However, the simplistic localization approach adopted in this study (sequential least squares) does not allow for a smooth position solution. It is suggested, however, that the results reported here have the potential of significant improvement using more advanced positioning algorithms (e.g., Kalman filter, particle filter or machine learning-based approaches) and additional-type observables, mainly embedded MEMS IMUs, while Wi-Fi RTT technology remains the core sensing information.

## Figures and Tables

**Figure 1 sensors-23-02829-f001:**
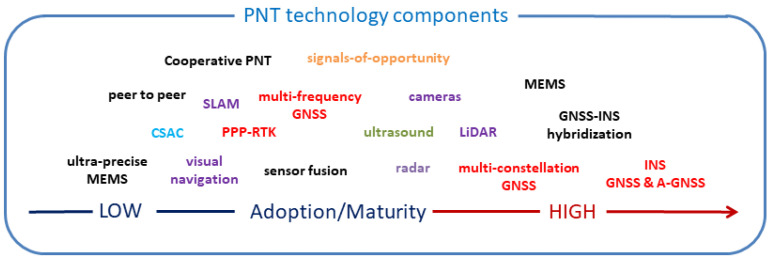
The PNT ecosystem indicating the state of maturity and adoption of the different sensors, technologies and techniques. This figure was made as in [34].

**Figure 2 sensors-23-02829-f002:**
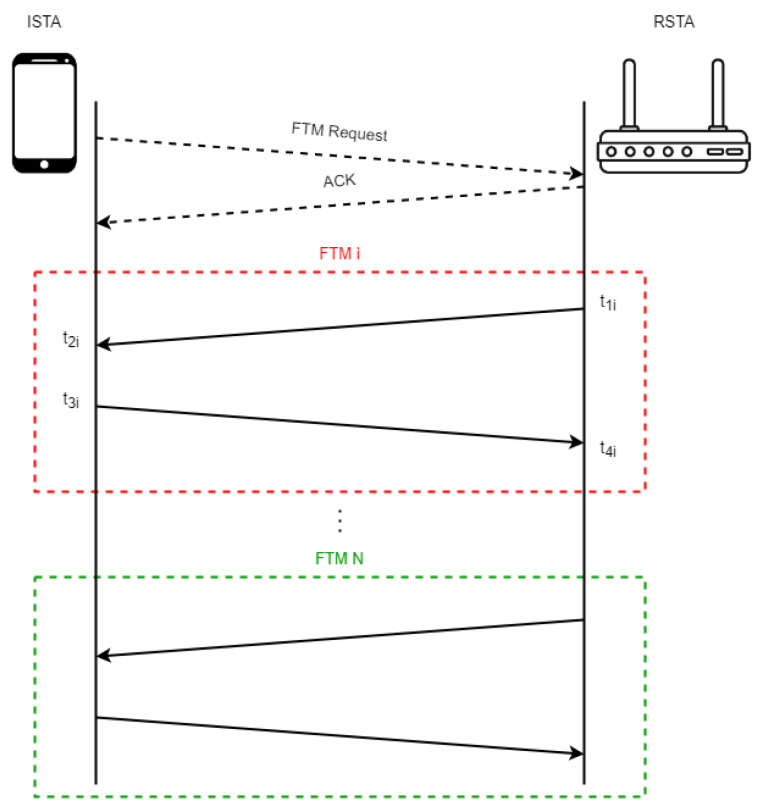
Operational principle of Wi-Fi RTT FTM.

**Figure 3 sensors-23-02829-f003:**
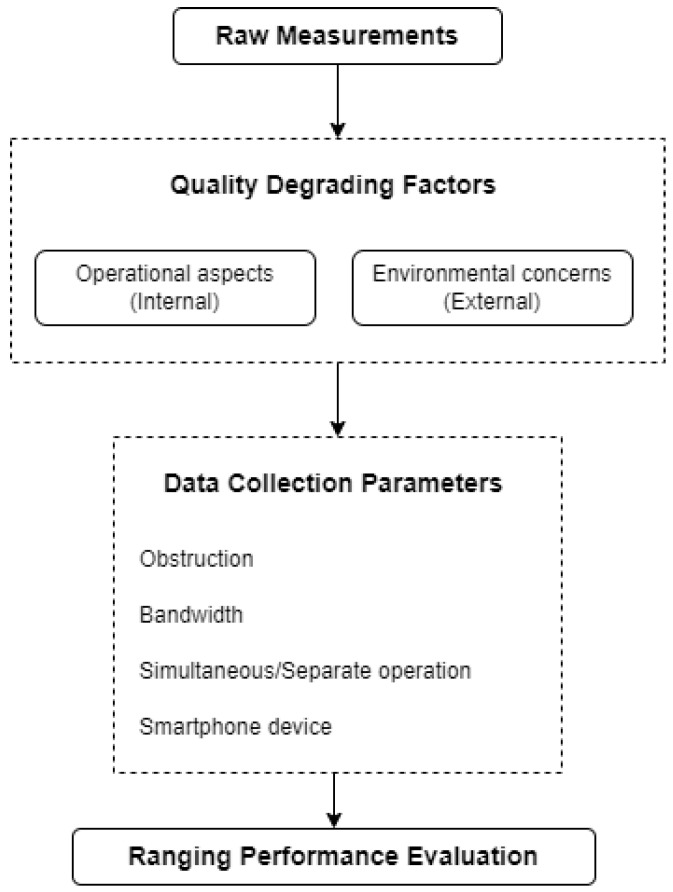
Process of ranging performance evaluation.

**Figure 4 sensors-23-02829-f004:**
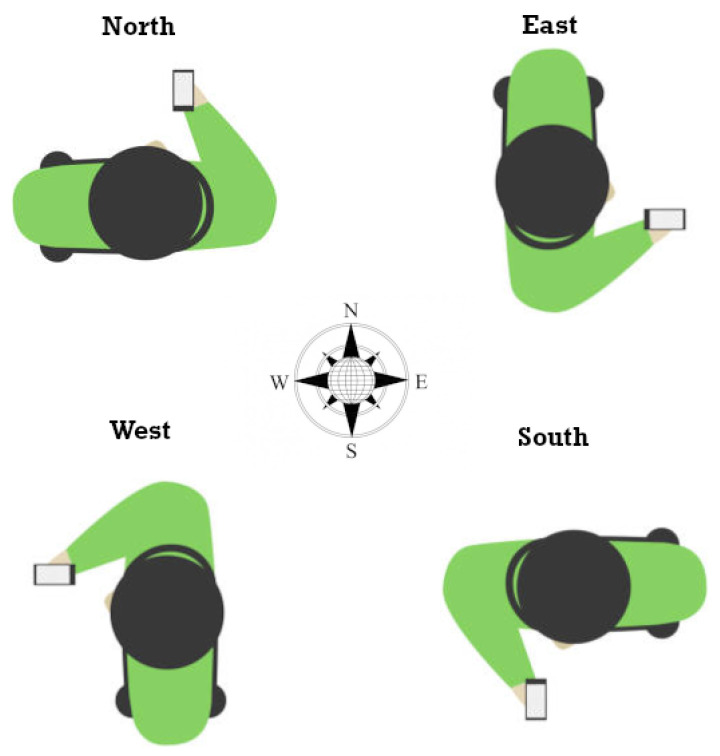
Representation of the four user orientations.

**Figure 5 sensors-23-02829-f005:**
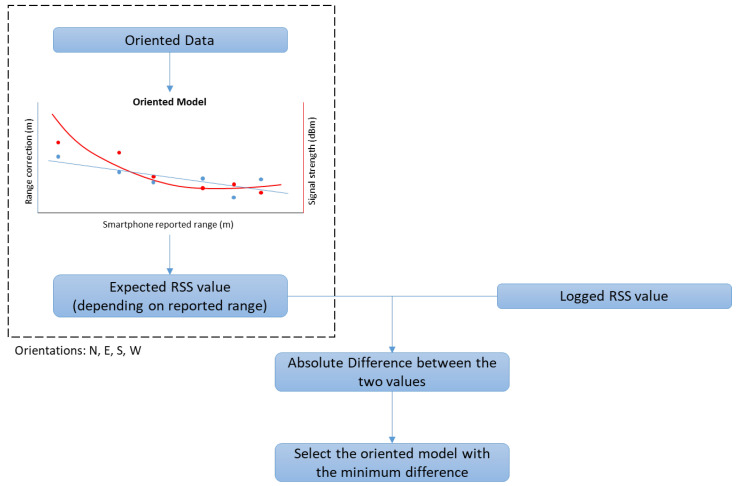
Process of the RSSI-based selection of the calibration model. The dotted box shows the procedure for each already known orientation model (×4). Red markers and line represent the signal strength data and polynomial fit respectively, cyan markers and line represent the ranging values and the linear calibration model respectively.

**Figure 6 sensors-23-02829-f006:**
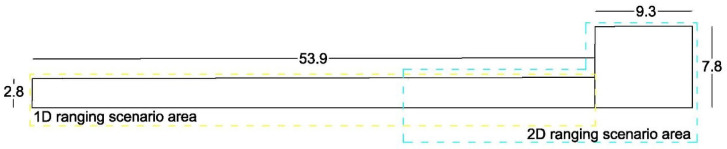
Map of the ground floor test area in a multi-story building. Dimensions are in meters (m).

**Figure 7 sensors-23-02829-f007:**
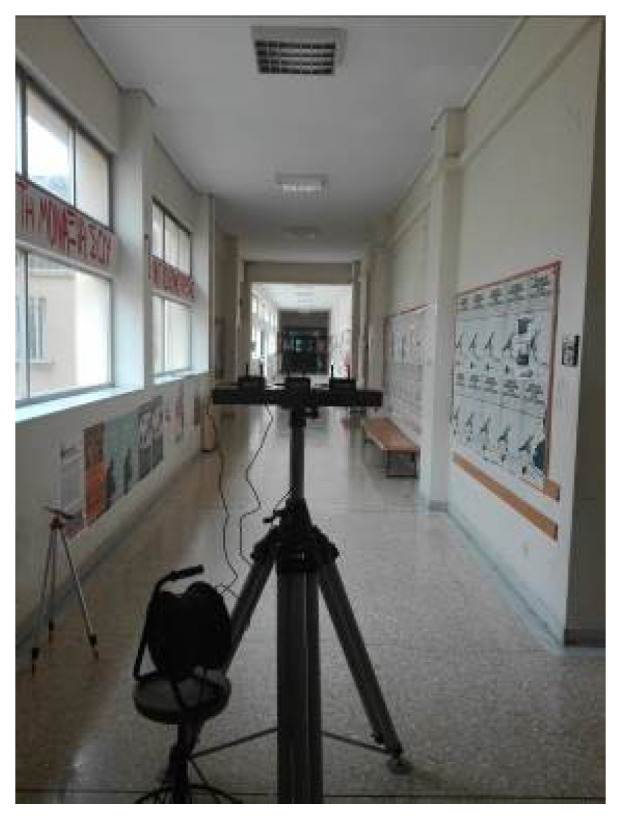
1D ranging corridor area.

**Figure 8 sensors-23-02829-f008:**
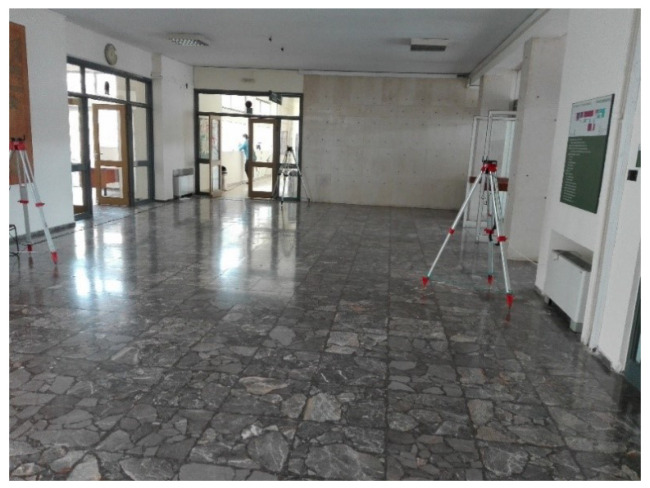
2D ranging lobby area.

**Figure 9 sensors-23-02829-f009:**
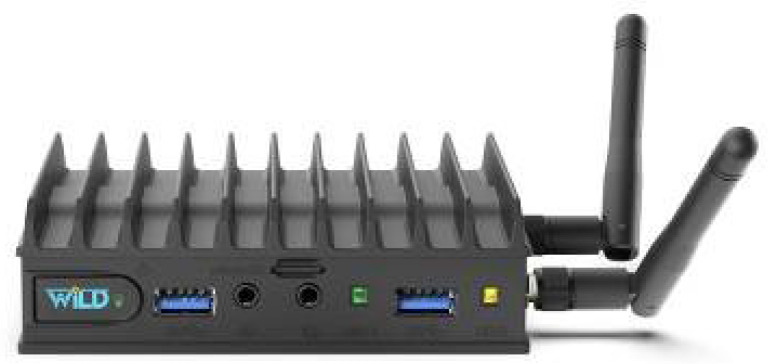
Compulab WILD access point. (https://fit-iot.com/web/products/wild/ (accessed on 23 February 2023)).

**Figure 10 sensors-23-02829-f010:**
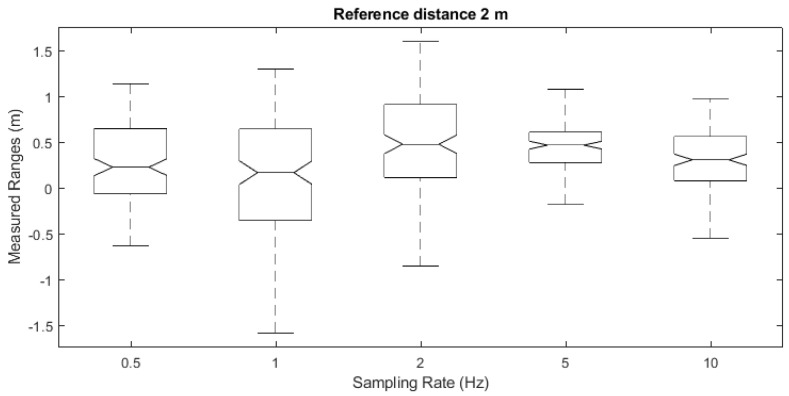
Wi-Fi RTT sampling rate statistics box plots.

**Figure 11 sensors-23-02829-f011:**
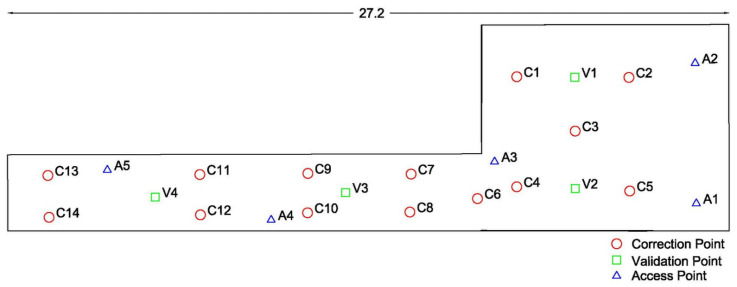
Spatial distribution of the adopted access, correction and validation points during the 2D test trials. Dimensions are in meters (m).

**Figure 12 sensors-23-02829-f012:**
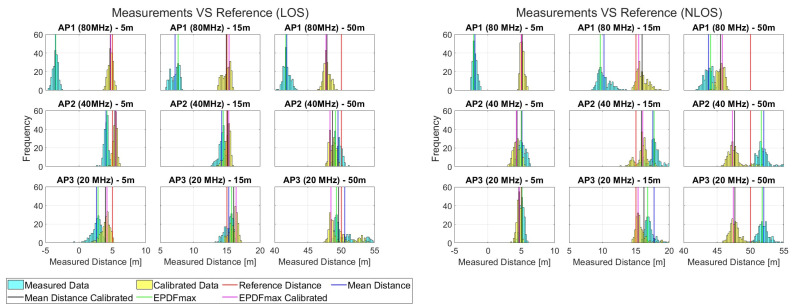
Histograms of ranges for the first two scenarios in the 1D experiment.

**Figure 13 sensors-23-02829-f013:**
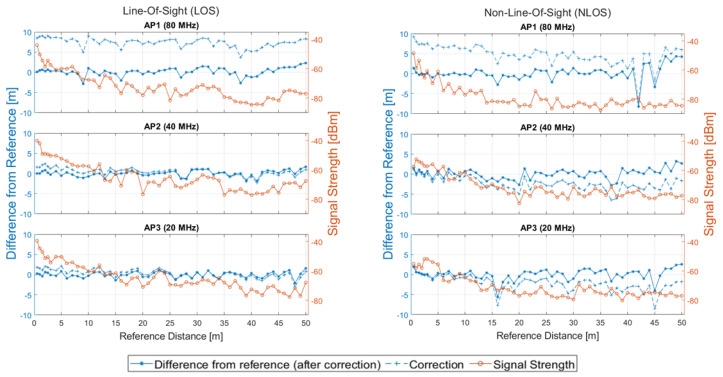
Ranging and correction performance of the first two scenarios in the 1D experiment.

**Figure 14 sensors-23-02829-f014:**
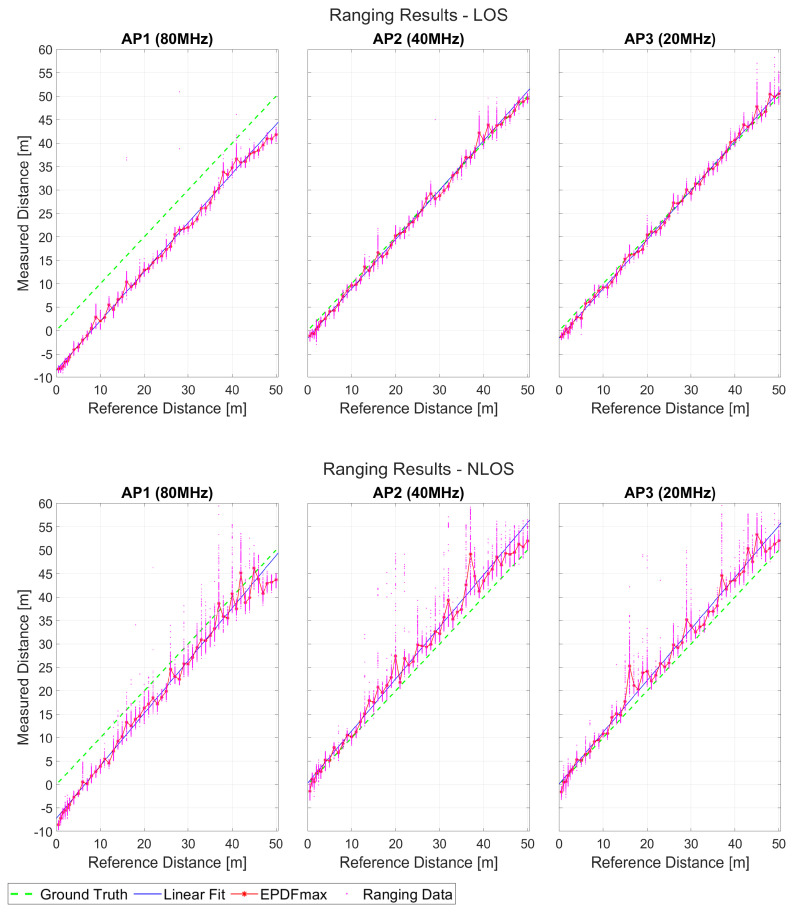
Comparison of the ranging results for the first two scenarios.

**Figure 15 sensors-23-02829-f015:**
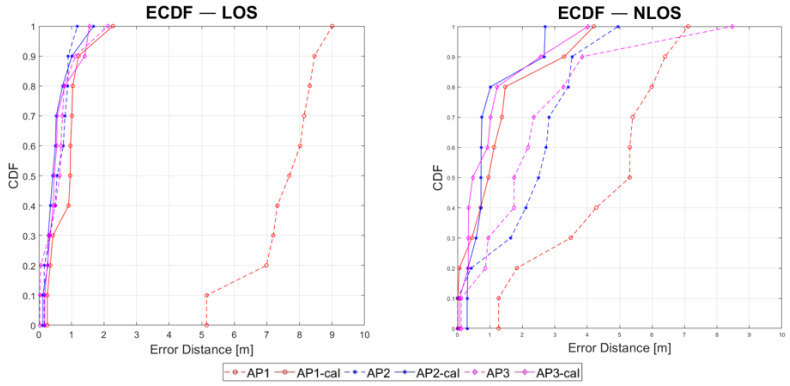
Empirical CDF for comparing the three APs and for identifying their respective performance change in LOS and NLOS conditions.

**Figure 16 sensors-23-02829-f016:**
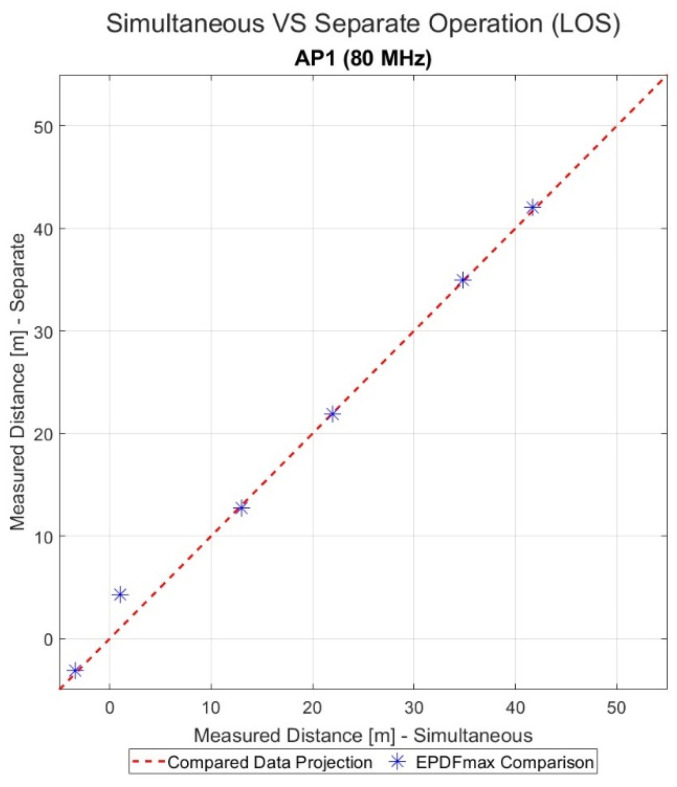
Comparison of EPDFmax values for AP1 for scenarios 1 (*x*-axis) and 3 (*y*-axis).

**Figure 17 sensors-23-02829-f017:**
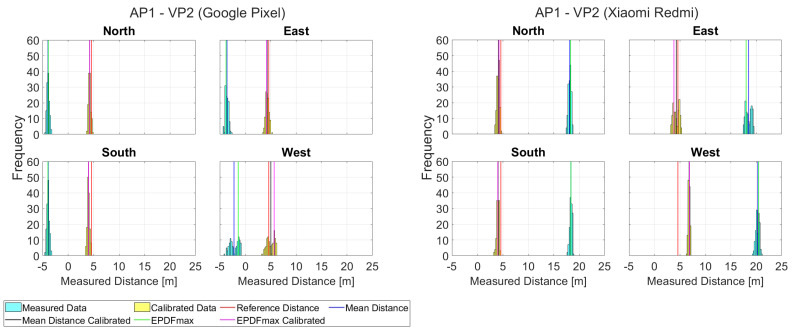
Histograms of ranges between AP1 and VP2, for different user orientations and smartphone devices.

**Figure 18 sensors-23-02829-f018:**
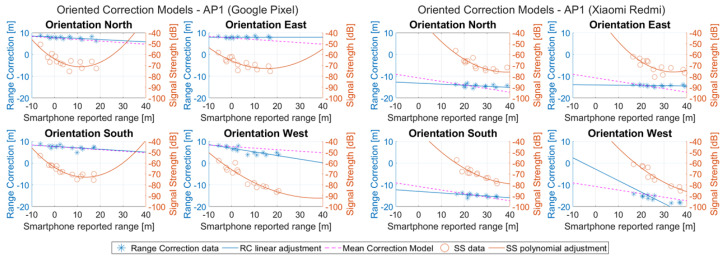
Range correction models and RSSI second-order polynomial curves of AP1, for different user orientations and smartphone devices.

**Figure 19 sensors-23-02829-f019:**
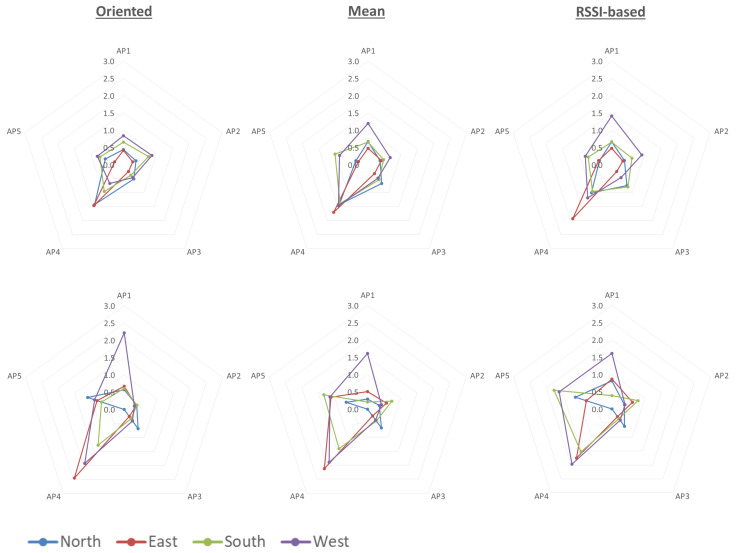
RMSE comparison between the different correction models for the complete dataset of VP2, for the different smartphone devices (Google Pixel—**top**, Xiaomi Redmi—**bottom**).

**Figure 20 sensors-23-02829-f020:**
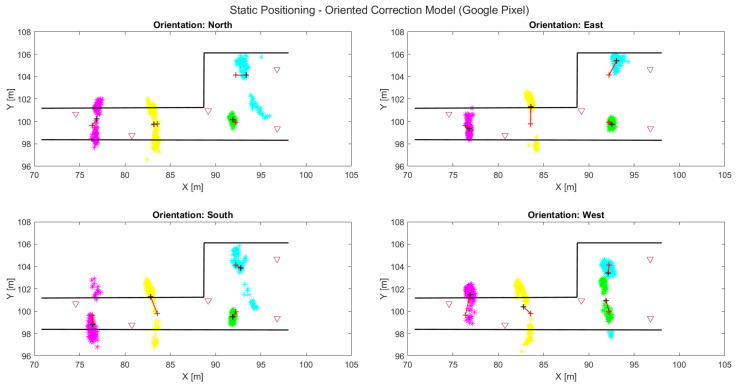
Comparison of static positioning by using the oriented correction model at all of the VPs, for the different oriented datasets and smartphone devices.

**Table 1 sensors-23-02829-t001:** Equipment employed during the experimental campaigns.

Campaign	1D	2D
**Number of Smartphones used**	1	2
**Smartphones**	Google Pixel 3a XL	Google Pixel 3a XL and Xiaomi Redmi Note 9 Pro
**Amount ofAccess Points**	3	5
**Access Points**	Compulab Wi-Fi Indoor Location Device (WILD)	Compulab Wi-Fi Indoor Location Device (WILD)

**Table 2 sensors-23-02829-t002:** 1D positioning: Scenarios configuration.

	Scenario 1	Scenario 2	Scenario 3
**AP Settings**	Bandwidth: AP1-80 MHz, AP2-40 MHz, AP3-20 MHz Sampling Rate: 10 Hz		
**AP operation**	3 APs simultaneously	3 APs simultaneously	Each AP separately
**Experiment Conditions**	LOS	NLOS	LOS
**Number of** **Reference Ranges**	53 RP (43 CP, 10 VP)	53 RP (43 CP, 10 VP)	6 RP
**Selection of** **Reference Ranges**	0.5–3.0 m per 0.5 m, 3–50 m per 1 m	0.5–3.0 m per 0.5 m, 3–50 m per 1 m	5, 10, 20, 30, 40, 50 m

**Table 3 sensors-23-02829-t003:** 2D positioning: Ranging measurements configuration.

2D Positioning—Correction (Ranging Measurements)	
**AP settings**	Bandwidth: 80 MHz, Sampling Rate: 10 Hz
**Number of APs**	5 Access Points
**CP**	14 Correction Points
**VP**	4 Validation Points

**Table 4 sensors-23-02829-t004:** Ranging comparison of simultaneous/separate operation between the different APs.

Absolute Difference (m)			
**RD (m)**	**AP1** **(80 MHz)**	**AP2** **(40 MHz)**	**AP3** **(20 MHz)**
**5**	0.36	0.27	1.13
**10**	3.31	0.89	0.14
**20**	0.28	1.03	0.29
**30**	0.06	0.41	0.21
**40**	0.11	0.38	2.61
**50**	0.40	0.47	3.91

**Table 5 sensors-23-02829-t005:** Parameters and goodness-of-fit metric of linear models for each oriented dataset. Smartphone device: Google Pixel.

Google Pixel								
	**North**		**East**		**South**		**West**	
	**(a, b)**	**SE (m)**	**(a, b)**	**SE (m)**	**(a, b)**	**SE (m)**	**(a, b)**	**SE (m)**
AP1	(−0.0537, 7.954)	0.62	(0.0003, 7.936)	0.42	(−0.0647, 7.657)	0.90	(−0.1676, 6.824)	1.03
AP2	(−0.1129, 6.838)	0.83	(−0.1425, 7.393)	0.70	(−0.1559, 7.461)	1.00	(−0.1248, 6.851)	1.14
AP3	(−0.0309, 7.929)	0.40	(−0.0708, 7.375)	1.01	(−0.0725, 7.851)	0.54	(−0.2129, 6.898)	0.84
AP4	(−0.3380, 7.343)	1.80	(−0.2561, 6.989)	1.22	(−0.2492, 7.404)	1.29	(−0.1573, 7.520)	0.85
AP5	(−0.2380, 8.044)	2.10	(−0.1970, 7.753)	1.24	(−0.1912, 8.000)	1.40	(−0.1839, 7.721)	1.50

**Table 6 sensors-23-02829-t006:** Parameters and goodness-of-fit metric of linear models for each oriented dataset. Smartphone device: Xiaomi Redmi.

Xiaomi Redmi								
	**North**		**East**		**South**		**West**	
	**(a, b)**	**SE (m)**	**(a, b)**	**SE (m)**	**(a, b)**	**SE (m)**	**(a, b)**	**SE (m)**
AP1	(−0.0482,−13.177)	0.76	(−0.0112,−14.006)	0.41	(−0.0728,−13.000)	0.58	(−0.5262,−2.794)	2.93
AP2	(−0.1161,−12.578)	0.87	(−0.1373,−11.850)	0.93	(−0.2000,−10.361)	0.96	(−0.1194,−13.037)	1.24
AP3	(−0.0309,−13.679)	0.46	(−0.0738,−12.940)	0.90	(−0.0931,−12.529)	0.39	(−0.1749,−11.366)	0.62
AP4	(−0.3478,−7.274)	1.92	(−0.2042,−10.942)	1.18	(−0.2780,−8.864)	1.57	(−0.2759,−9.089)	1.33
AP5	(−0.2304,−9.261)	1.55	(−0.2049,−10.081)	1.30	(−0.1333,−11.308)	1.04	(−0.1443,−11.356)	1.45

**Table 7 sensors-23-02829-t007:** Parameters and goodness-of-fit metric of second-order polynomial models for each oriented dataset. Smartphone device: Google Pixel.

Google Pixel				
	**North**		**East**	
	**(a, b, c)**	**SE (dBm)**	**(a, b, c)**	**SE (dBm)**
AP1	(0.055, −1.251,−64.2)	4.4	(0.030, −0.922, −65.9)	3.5
AP2	(0.044, −1.904,−64.0)	3.2	(0.026, −1.588, −66.2)	3.8
AP3	(0.086, −1.037,−67.9)	3.9	(0.055, −1.571, −68.6)	4.6
AP4	(−0.025,−1.553,−67.8)	6.1	(0.023, −2.095, −70.2)	4.5
AP5	(−0.010,−1.260,−65.5)	4.4	(−0.042,−0.865, −68.2)	3.1
	**South**		**West**	
	**(a, b, c)**	**SE (dBm)**	**(a, b, c)**	**SE (dBm)**
AP1	(0.049, −1.318,−64.0)	2.2	(0.018, −1.385,−66.0)	3.9
AP2	(0.023, −1.666,−63.8)	3.5	(0.019, −1.633,−67.2)	3.6
AP3	(0.026, −1.361,−68.6)	4.7	(0.067, −2.031,−71.8)	3.2
AP4	(−0.019,−1.675,−67.5)	5.7	(−0.029,−1.688,−70.0)	3.8
AP5	(−0.053,−0.895,−62.9)	5.6	(−0.015,−1.182,−66.7)	4.9

**Table 8 sensors-23-02829-t008:** Parameters and goodness-of-fit metric of second-order polynomial models for each oriented dataset. Smartphone device: Xiaomi Redmi.

Xiaomi Redmi				
	**North**		**East**	
	**(a, b, c)**	**SE (dBm)**	**(a, b, c)**	**SE (dBm)**
AP1	(−0.0482,−13.177, 0.76)	3.3	(−0.0112,−14.006, 0.41)	4.4
AP2	(−0.1161,−12.578, 0.87)	3.6	(−0.1373,−11.850, 0.93)	3.8
AP3	(−0.0309,−13.679, 0.46)	4.6	(−0.0738,−12.940, 0.90)	5.2
AP4	(−0.3478,−7.274, 1.92)	5.5	(−0.2042,−10.942, 1.18)	4.3
AP5	(−0.2304,−9.261, 1.55)	3.6	(−0.2049,−10.081, 1.30)	2.8
	**South**		**West**	
	**(a, b, c)**	**SE (dBm)**	**(a, b, c)**	**SE (dBm)**
AP1	(−0.0728,−13.000, 0.58)	2.7	(−0.5262,−2.794, 2.93)	3.3
AP2	(−0.2000,−10.361, 0.96)	2.4	(−0.1194,−13.037, 1.24)	3.8
AP3	(−0.0931,−12.529, 0.39)	3.6	(−0.1749,−11.366, 0.62)	6.7
AP4	(−0.2780,−8.864, 1.57)	3.2	(−0.2759, −9.089, 1.33)	3.3
AP5	(−0.1333,−11.308, 1.04)	5.9	(−0.1443,−11.356, 1.45)	4.3

**Table 9 sensors-23-02829-t009:** RSSI-based oriented model selection performance of the Google Pixel device.

Google Pixel								
	**N**	**E**	**S**	**W**	**N**	**E**	**S**	**W**
	**% of Correct Model Selection**				**RMSE (m) (STD (m))**			
**VP1**								
**AP1**	0.8	72.4	0.0	80.2	1.12 (1.03)	1.45 (0.79)	0.86 (0.86)	0.76 (0.75)
**AP2**	16.9	98.4	9.8	0.8	0.32 (0.31)	0.59 (0.52)	0.48 (0.47)	3.01 (0.89)
**AP3**	0.0	0.0	8.2	0.0	0.46 (0.28)	0.94 (0.51)	0.47 (0.47)	0.71 (0.32)
**AP4**	0.0	4.1	0.0	72.7	2.42 (1.55)	2.92 (0.65)	2.32 (0.83)	1.99 (1.13)
**AP5**	24.2	6.5	44.3	0.0	2.92 (0.76)	3.16 (1.34)	0.99 (0.86)	1.10 (0.41)
**VP2**								
**AP1**	5.6	78.4	100.0	22.9	0.67 (0.25)	0.48 (0.38)	0.66 (0.23)	1.19 (0.74)
**AP2**	86.3	0.9	0.0	1.0	0.36 (0.27)	0.41 (0.19)	0.46 (0.30)	0.67 (0.67)
**AP3**	1.6	3.4	0.7	45.7	0.67 (0.47)	0.31 (0.19)	0.54 (0.22)	0.48 (0.42)
**AP4**	100.0	12.1	0.0	9.5	1.42 (0.28)	1.70 (0.81)	1.38 (0.16)	1.46 (0.27)
**AP5**	14.5	20.7	0.0	60.0	0.38 (0.32)	0.30 (0.29)	1.01 (0.22)	0.87 (0.78)
**VP3**								
**AP1**	0.0	0.0	5.8	100.0	0.72 (0.34)	1.67 (0.68)	1.35 (0.86)	1.37 (0.55)
**AP2**	5.2	0.0	2.4	29.9	2.70 (1.38)	0.78 (0.74)	1.51 (0.64)	2.30 (1.06)
**AP3**	28.1	0.0	0.0	29.9	0.71 (0.28)	0.30 (0.26)	0.42 (0.29)	1.35 (0.78)
**AP4**	97.8	91.3	18.3	6.8	0.79 (0.20)	1.65 (0.22)	1.29 (0.30)	0.83 (0.30)
**AP5**	18.5	5.2	0.0	0.0	0.82 (0.37)	0.50 (0.38)	0.72 (0.22)	0.51 (0.51)
**VP4**								
**AP1**	0.0	0.0	7.3	100.0	0.74 (0.36)	3.51 (0.46)	0.55 (0.52)	1.19 (0.54)
**AP2**	10.7	91.2	20.3	0.0	1.10 (0.78)	0.62 (0.30)	1.26 (1.05)	0.81 (0.36)
**AP3**	0.0	0.0	8.9	100.0	1.37 (0.36)	0.70 (0.37)	0.90 (0.84)	0.64 (0.52)
**AP4**	6.6	64.0	0.0	0.0	0.70 (0.24)	0.44 (0.19)	0.48 (0.41)	0.82 (0.39)
**AP5**	93.4	0.0	0.0	24.8	0.74 (0.36)	1.17 (0.45)	0.79 (0.65)	0.68 (0.24)

**Table 10 sensors-23-02829-t010:** RSSI-based oriented model selection performance of the Xiaomi Redmi device (*NaN*: not available dataset).

Xiaomi Redmi								
	**N**	**E**	**S**	**W**	**N**	**E**	**S**	**W**
	**% of Correct Model Selection**				**RMSE (m) (STD (m))**			
**VP1**								
**AP1**	30.0	0.0	3.3	21.9	1.07 (0.43)	0.59 (0.24)	0.61 (0.27)	0.75 (0.41)
**AP2**	100.0	0.0	0.0	0.0	0.96 (0.24)	0.66 (0.39)	0.32 (0.32)	4.85 (0.63)
**AP3**	91.7	0.0	42.1	4.4	0.32 (0.31)	0.37 (0.26)	1.14 (0.55)	0.34 (0.24)
**AP4**	100.0	0.0	0.0	0.0	0.96 (0.96)	0.61 (0.25)	0.52 (0.52)	0.99 (0.53)
**AP5**	100.0	0.0	0.0	0.0	1.48 (0.84)	5.12 (2.69)	1.64 (0.44)	0.54 (0.38)
**VP2**								
**AP1**	0.0	0.0	4.8	15.2	0.81 (0.26)	0.87 (0.67)	0.39 (0.28)	1.61 (0.45)
**AP2**	100.0	0.0	0.0	0.0	0.38 (0.35)	0.63 (0.22)	0.79 (0.34)	0.40 (0.40)
**AP3**	5.0	0.0	98.4	0.0	0.61 (0.47)	0.27 (0.25)	0.34 (0.19)	0.40 (0.40)
**AP4**	*NaN*	0.0	0.0	0.0	*NaN*	1.75 (1.37)	1.53 (0.26)	1.97 (0.23)
**AP5**	100.0	0.0	0.0	0.0	1.11 (0.36)	0.78 (0.77)	0.76 (0.26)	1.61 (0.27)
**VP3**								
**AP1**	*NaN*	22.7	0.8	75.4	*NaN*	1.55 (1.44)	1.43 (0.41)	2.52 (2.45)
**AP2**	5.1	*NaN*	0.0	0.0	1.06 (0.54)	*NaN*	1.22 (0.98)	2.24 (0.75)
**AP3**	100.0	0.0	0.0	0.0	0.59 (0.21)	0.48 (0.41)	0.89 (0.51)	0.72 (0.39)
**AP4**	100.0	0.0	0.0	0.0	1.50 (0.44)	0.97 (0.45)	0.90 (0.14)	1.10 (0.11)
**AP5**	100.0	0.0	0.0	0.0	1.04 (1.04)	0.29 (0.22)	0.63 (0.20)	0.99 (0.97)
**VP4**								
**AP1**	*NaN*	20.0	0.0	76.8	*NaN*	0.90 (0.37)	0.81 (0.47)	4.54 (4.02)
**AP2**	29.7	0.0	0.0	0.0	0.85 (0.64)	1.83 (0.37)	0.56 (0.52)	1.49 (0.57)
**AP3**	86.7	0.0	0.0	92.9	0.73 (0.58)	1.03 (0.95)	0.76 (0.57)	0.88 (0.47)
**AP4**	100.0	0.0	0.0	0.0	0.20 (0.20)	0.35 (0.23)	0.59 (0.43)	0.90 (0.50)
**AP5**	100.0	0.0	0.0	37.4	0.77 (0.22)	0.91 (0.23)	0.93 (0.21)	0.85 (0.33)

## Data Availability

Data available on request.

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
