# Peer review of "Testing and Evaluation of Wi-Fi RTT Ranging Technology for Personal Mobility Applications"

_sensors, 2023, doi:10.3390/s23052829_

Round 1
Reviewer 1 Report
Dear Authors,
Please find the attached file for your reference. Please update the paper based on the comments and resubmit it.
Regards

Author Response
Dear Reviewer,
please find our reply to your comments in the attached document.
Many thanks for your time to review.
Best regards,
Guenther Retscher

Reviewer 2 Report
1. Some comparisons other than RTT should be presented to show advantage or disadvantage of RTT.
2. More theoretical parts about RTT should be analyzed and given.
3. If more advanced positioning algorithms are further used, how much improvement can be obtained? It is interesting to show some results in this regard.
Author Response

(The authors gave the same response as above.)

Reviewer 3 Report
1. I suggest the author to highlight the contribution of the paper. It is not clear the exact problem addressed in this paper. It is a localization problem or it is just a focus of the ranging?
2. I suggest the authors adding a section called related work to talk about the related works in this area.
3. Only testing might not be enough for publication at the journal. The authors should highlight the methodology portion.
4. Is there any relationship between RSS and the WiFi RTT, as we all know that the RSS is highly impacted by the environment. How about the RTT if there are occlusions in the environment? The authors should add discussion on this point.
5. What is the difference between 1D ranging and 2D ranging? Will the setup impact the RTT accuracy?
6. Is is not easy to see any useful information in Figure 10. I suggest to remove this figure.
7. In figure 12, how you determine the correction point? Are you doing 2D localization for that?
8. In figure 16, it seams the LOS gives much more error than NLOS. Please explain why.
9. In table 5, it is not clear what do you mean by “% of correct model section” .
Author Response

(The authors gave the same response as above.)

Reviewer 4 Report
This study aims to investigate the performance of ranging in different scenarios of Wi-Fi RTT Ranging Technology such as LOS/NLOS, Bandwith and smartphone devices of other brands and propose a correction model. The relevant opinions are as follows:
1. The article makes a reasonable explanation for indoor positioning, especially wifi technology.
2. This article's smartphone device and Aps are the evaluation objects for wifi ranging. It is recommended to explain the relevant essential specifications for the table or figure of the equipment used.
3. The scenarios in the study are divided into 1D and 2D. In the 1D, 80MHz, 40MHz, and 20MHz Bandwidths are used, and the authors said that 80MHz has a better ranging performance. Therefore, only 80MHz is used for the test in the 2D scenario. However, in the 2D scenario, which Bandwidth performs better has not been tested in the research.
4. For the display of Fig. 10, please refer to ''Martin-Escalona, I., & Zola, E. (2023). Improving Fingerprint-Based Positioning by Using IEEE 802.11 mc FTM/RTT Observables. Sensors,23(1), 267 .'' for better readability.
5. Fig 12 suggests adding a distance label.
6. Fig13 and 18 are difficult to see what the authors want to express. They are blurry.
7. The ranging correction model is one of the main purposes of this research. It is recommended to add model parameters and accuracy indicators.
8. Table 4 and Table 5 only use a single accuracy indicator, and it is recommended to add such as MSE, RMSE, MAE, STD, ME, Accuracy(%), etc., to strengthen the research results.
9. Fig.21 suggests that the error ellipse can be used to display the positioning accuracy.
10. To show the results of this article, it is suggested to refer to the display types of the following articles to make it easier for readers to understand the content of the findings:
Martin-Escalona, I., & Zola, E. (2023). Improving Fingerprint-Based Positioning by Using IEEE 802.11 mc FTM/RTT Observables. Sensors, 23(1), 267.
Ando, H., Sekoguchi, S., Ikegami, K., Yoshitake, H., Baba, H., Myojo, T., & Ogami, A. (2021). Combining indoor positioning using Wi-Fi round trip time with dust measurement in the field of occupational health. ensors, 1(21), 7261.
Author Response

(The authors gave the same response as above.)

Round 2
Reviewer 1 Report
Dear Authors,
Thank you for addressing all my comments and I don't have any further concerns about your paper.
Regards
Author Response
Dear Reviewer,
thank you very much for accepting the revised version of our paper.
Best regards,
Guenther Retscher
Reviewer 2 Report
No more further comment
Author Response

(The authors gave the same response as above.)

Reviewer 3 Report
The text in some figures (for example Figure 19) are hard to see. Also some figures (for example Figure 20) do not have any legends.
Author Response
Dear Reviewer,
Thank you for comment.
We have modified Figure 19 for better readability.
Figure 14 had the legend between the Figures in the middle. This is now changed and the legend is in the bottom of the Figure.
Figure 20 has now a legend also in the bottom.
For final typesetting, such as placement of Figure 19 over the whole page we are asking the typesetters to assist us.
Best regards,
Guenther Retscher
Reviewer 4 Report
No more questions!
Author Response

(The authors gave the same response as above.)
